# Livestock Informatics Toolkit: A Case Study in Visually Characterizing Complex Behavioral Patterns across Multiple Sensor Platforms, Using Novel Unsupervised Machine Learning and Information Theoretic Approaches

**DOI:** 10.3390/s22010001

**Published:** 2021-12-21

**Authors:** Catherine McVey, Fushing Hsieh, Diego Manriquez, Pablo Pinedo, Kristina Horback

**Affiliations:** 1Department of Animal Science, University of California Davis, Davis, CA 95616, USA; kmhorback@ucdavis.edu; 2Department of Statistics, University of California Davis, Davis, CA 95616, USA; fhsieh@ucdavis.edu; 3Department of Animal Science, Colorado State University, Fort Collins, CO 80523, USA; diego.manriquez_alvarez@colostate.edu (D.M.); pablo.pinedo@colostate.edu (P.P.)

**Keywords:** dairy welfare, hierarchical clustering, mutual information, precision livestock farming, time budgets, unsupervised machine learning

## Abstract

Large and densely sampled sensor datasets can contain a range of complex stochastic structures that are difficult to accommodate in conventional linear models. This can confound attempts to build a more complete picture of an animal’s behavior by aggregating information across multiple asynchronous sensor platforms. The Livestock Informatics Toolkit (LIT) has been developed in R to better facilitate knowledge discovery of complex behavioral patterns across Precision Livestock Farming (PLF) data streams using novel unsupervised machine learning and information theoretic approaches. The utility of this analytical pipeline is demonstrated using data from a 6-month feed trial conducted on a closed herd of 185 mix-parity organic dairy cows. Insights into the tradeoffs between behaviors in time budgets acquired from ear tag accelerometer records were improved by augmenting conventional hierarchical clustering techniques with a novel simulation-based approach designed to mimic the complex error structures of sensor data. These simulations were then repurposed to compress the information in this data stream into robust empirically-determined encodings using a novel pruning algorithm. Nonparametric and semiparametric tests using mutual and pointwise information subsequently revealed complex nonlinear associations between encodings of overall time budgets and the order that cows entered the parlor to be milked.

## 1. Introduction

Precision livestock farming (PLF) technologies produce prodigious amounts of data [1]. Although the behaviors encoded by such sensors are often much simpler than those that can be quantified by a human observer, the measurement granularity and perseverance provided by these technologies creates new opportunities to study complex behavioral patterns across time and in a wider range of contexts. Observations collected on a single animal over extended observation windows at high sampling frequencies can, however, contain a range of complex temporal patterns, such as cyclicity, non-stationarity, autocorrelation, etc. [2]. Furthermore, when sensors are applied to large heterogenous groups of animals housed socially in spatially restricted environments, recorded behaviors may also contain complex interdependencies between animals at the dyadic, triadic, clique, and herd levels [3,4,5]. Failing to accommodate all these complex structural and stochastic features in a conventional model-based approach to statistical inference risks returning spurious insights into the underlying behavioral dynamics. Developing such a model with a single PLF data stream can be challenging. Provided multiple data streams, however, the logistical challenges presented by model-based analytical frameworks can rapidly compound, creating significant barriers to cross-sensor inferences, and thereby impeding researchers from extracting more holistic behavioral inferences from increasingly data-rich farm environments.

Unsupervised Machine Learning (UML) tools may provide a more flexible and forgiving approach to knowledge discovery in the context of large sensor datasets [6,7]. Such algorithms excel at identifying and characterizing complex non-random behavioral patterns lying beneath the stochastic surface of a dataset, while often employing relatively few structural assumptions about the data [8,9,10]. Hierarchical clustering-based techniques offer an intuitive and highly adaptable approach to visualizing high dimensional datasets that is particularly well-suited to exploratory data analysis [4,9]. Indeed, by reducing the complex behavioral signals present in a sensor dataset into a series of discrete clusters, such algorithms may be viewed as an empirical extension of classical ethological techniques. Discrete data, however, can be challenging to work with in most frequentist and even many Bayesian frameworks. Estimators based on information entropy, on the other hand, are purpose-made to quantify uncertainty in discretely encoded data without knowledge of the underlying distribution, and thus naturally complement hierarchical clustering-based algorithms [7,11,12].

Clustering algorithms, by virtue of their incredible flexibility, have successfully been applied to a range of PLF data streams [7,13,14,15,16,17,18]. In our own previous work, we have highlighted the utility of hierarchical clustering-based approaches in leveraging the behavioral co-dependencies of cows housed socially in large groups, in a production environment, in order to recover complex temporal patterns in behavior [7]. In these analyses, data mechanics algorithms were able to recover complex nonstationarity in the order in which cows entered the milking parlor. Some of these changes in queuing patterns could be attributed to the shift to spring pasture access, but other transient and persistent shifts in entry order recovered in these encodings were driven by environmental factors not experimentally recorded [7,19,20]. Entropy-based nonparametric permutation tests were also successful in recovering preliminary evidence of significant nonlinear associations between encodings of entry-order patterns and activity patterns recorded using ear-tag accelerometers. In this paper we will explore how novel ensemble simulation techniques [11] that emulate and adjust for the complex sources of error in PLF data streams may be used to produced more balanced encodings of multi-dimensional behavioral data. We also introduce a new dendrogram pruning algorithm that is able to efficiently repurpose these same ensemble simulations, to ensure that that the power of hierarchical clustering tools do not exceed the resolution of the sensor. Finally, we demonstrate the utility of information decomposition techniques within our existing nonparametric mutual information testing framework, to better facilitate the visual characterization of complex behavioral patterns across sensor data sets that might be overlooked in more conventional model-based analyses.

## 2. Materials and Methods

### 2.1. Description of Data

To demonstrate the efficacy of our analytical approach, data was repurposed from a feed trial assessing the impact of an organic fat supplement on cow health and productivity, through the first 150 days of lactation. All animal handling and experimental protocols were approved by the Colorado State University Institution of Animal Care and Use Committee (Protocol ID: 16-6704AA). The study ran from January through July in 2017, on a USDA Certified Organic dairy in Northern Colorado, enrolling a total of 200 cows over a 1.5-month period into a mixed-parity herd of animals, with predominantly Holstein genetics. Cows were maintained in a closed herd in an open-sided free-stall barn, stocked at roughly half capacity with respect to both feed bunk spaces and stalls. Cows had free access to an adjacent outdoor dry lot while in their home pen, and beginning in April were moved onto pasture at night, to comply with organic grazing standards. Cows were milked three times a day, with free access to TMR between milkings, and were head locked each morning to facilitate data collection and daily health checks. For more details on feed trial protocols, see Manriquez et al. (2018) and Manriquez et al. (2019) [21,22].

In addition to standard production and health assessments, behavioral data was also obtained from several PLF data streams [19]. Milking order, or the sequence in which cows enter the parlor to be milked, is automatically recorded as metadata in all modern RFID-equipped milking systems. Our study cows were milked in a DelPro^TM^ rotary parlor (DeLaval, Tumba, Sweden). At each morning milking, raw milking logs were exported from the parlor software, and the data were processed in order to extract the single-file order that cows entered the rotary [23]. A total of 80 milk order records—26 recorded while cows remained overnight in a free-stall barn, and 54 following the transition to overnight access to spring pasture—were used to create discrete encodings for parlor entry patterns via data mechanics clustering (see McVey et al. for further analytical details) [7]. The dendrograms summarizing the distribution of cow entry-order patterns and subsequent heatmap visualizations will be subjected to further analysis, without modifications to the previously reported encodings.

Animals enrolled in this feed trail were also fitted with a CowManager ear tag accelerometer (Agis Automatisering BV, Harmelen, The Netherlands). This commercial sensor platform, while designed and optimized for disease and heat detection, also provides hourly time budget estimates for total time (min) engaged in five mutually exclusive discrete behaviors—eating, rumination, non-activity, activity, and high activity [24,25]. Time budget data was collected on all animals for a contiguous period of 65 days (1560 h). The observation window began on 17 February, shortly after trial enrollment was completed, and ended on 23 April, when the grazing season commenced and cows were moved overnight beyond the range of the receiver antennae. After eliminating the data of cows that were removed prematurely from the observation herd due to acute clinical illness, as well as several cows with persistent receiver failure, complete sensor records were available for 179 animals. In order to focus fully on the logistical challenges of encoding and characterizing the complex multivariate dynamics of this system, we have chosen to compress this data over the time axis to consider only the overall time budgets of these cows, and will leave explorations of the longitudinal and cyclical complexity of this dataset for future work.

### 2.2. Improving Empirical Encodings of Overall Time Budget through Simulation

Regardless of its original distribution, data can always be coarsened into a discrete variable [26]. For complex or poorly defined systems, where appropriate cutoffs (binning rules) cannot be inferred *a priori*, an empirically-determined encoding may provide a more flexible and comprehensive approach to discretizing the underlying behavioral signals. One algorithm that provides a model-free approach to pattern encoding within the larger cannon of UML tools is hierarchical clustering. This approach employs a bottom-up agglomerative strategy to group observational units into discrete clusters of variable sizes, progressively building a coherent picture of herd-level global structures from the similarities in behavioral patterns observed between pairs of individuals [9,10]. This series of progressive pairings can be expressed graphically in the form of a dendrogram, which serves as a 2D representation of the data’s geometric distribution in its higher dimensional measurement space, and can subsequently be used in data visualizations to highlight the most prominent structural features of a dataset [19].

The efficacy of any hierarchical clustering scheme, however, is largely contingent on the adequacy of the estimator used to quantitatively express the pairwise dissimilarity between observational units [10]. The Euclidean distance (L2 norm) is the default estimator used in most applications of this algorithm [9,10,27], including much of the previous work in precision livestock applications [13,14,18]. The L2 norm is appropriate for many measurement systems where variance is reasonably uniform across a continuous domain of support. Time budget data, however, is distributed multinomially, and as such has significant domain constraints [26]. Put more simply, we know that the minutes logged for each behavior must sum to an hour. So, if a cow has ruminated for 60 min, then there can be no uncertainty in the remaining axes, because we know these values must be zero. These domain constraints impose co-dependencies between the behavioral axes that become stronger as observations shift towards the boundaries of the distribution’s support, which in turn warp the intrinsic variability of each axis contingent upon their location within the domain.

This statistical tedium also has some intuitive behavioral implications. Suppose we have two cows, Betty and Bessy, who spend 13 and 14 h a day ruminating, respectively. How “different” are these values? Since both cows are exceeding rumination rates needed to sustain a healthy metabolism, we would not anticipate that this difference would have a significant biological impact on these animals, and may ultimately be explained by relatively trivial behavioral fluctuations. Now, suppose instead that we have two other cows, Daisy and Delilah, who spend only 3 and 4 h a day ruminating, respectively. Given that both these cows are now well below the normal threshold for this behavior, this one-hour difference may have significant biological impacts. With a simple L2 norm, however, these two pairs would be given equivalent dissimilarity estimates for this behavioral axis, and so clearly a better estimator is needed.

Relative entropy, also referred to as the Kullback–Leibler divergence, is a classic information theoretic metric specifically designed to contrast discrete probability distributions, and thus a natural candidate for analysis of time budget data [12]. For any two distributions that utilize the same alphabet of k = 1 … K categorical features (i.e.,—use the same ethogram), relative entropy can be calculated using Equation (1), and converted to a symmetric distance measure using Equation (2). By utilizing the proportion of time that an animal invests in each behavior as both a nominal and relative value, this estimator is able to adjust the relative difference between cows by the absolute position of each observation relative to the boundary of the domain.
(1)$$D_{KL}(P||Q)=\overset{\;}{\underset{k}{\sum }}P\left(k\right)\;log\frac{P\left(k\right)}{Q\left(k\right)}\;\;$$
P=normalized time budget vector for Cow A 
Q=normalized time budget vector for Cow B 
(2)$$D_{KL}\left(P,Q\right)=D_{KL}(P||Q)+D_{KL}(Q||P)\;$$

Domain constraints are not, however, the only stochastic feature that need be accommodated when working with time budget data. There is also the measurement error attributable to the sensor itself. Returning to the previous example, suppose that we also know that our rumination records are only accurate to ±1 h. Is it then still appropriate to give more weight to the one-hour difference between Daisy and Delilah, than between Betty and Betsy? Since both observations are within the bounds of error, attempting to enhance the underlying biological signal may only succeed in amplifying measurement noise. A closed-form estimator, however, may not be readily generalizable to the wide range of measurement error models encountered with PLF sensors. We therefore propose that a simulation-based approach may offer a more flexible means of accounting for measurement errors in dissimilarity estimates [11].

The LIT package provides a built-in simulation utility for time budget data that seeks to mimic the stochastic error structure of the original data while still preserving the underlying behavioral signal [11]. Data is provided as a tensor, with cow indexed on the first axis, time indexed on the second, and the component behaviors on the final axis. The count data at each cow-by-time index is then used to redraw a simulated datapoint from one of three optional distributions [26]. In the first, the user may sample directly from a multinomial distribution centered around the normalized observed count vector. This model assumes that measurement error should shrink as a cow dedicates larger proportions of an observation window to specific behaviors, and intrinsically prevents estimates from being generated outside the domain of support. Variance can be underestimated at the extremes of the domain, however, if the probability for a behavior is non-negligible, but the observed count is zero due to under-sampling. This issue may be addressed in sampling option two, where samples are redrawn from a multivariate beta distribution (MBD), also known as a Dirichlet distribution, again parameterized using the normalized observed count. While this sampling strategy slightly biases the simulation towards the center of the distribution, it prevents under-sampling at the extremes of the domain. Finally, users may combine these sampling strategies in sampling option three, wherein the probability vector used to parameterize the multinomial is drawn first from the Dirichlet, in order to further increase the uncertainty in the simulated data. After simulation has been completed by redrawing samples at the finest level of temporal granularity supported by the sensor, the data can then be conditionally or fully aggregated along temporal axis as required for downstream analysis as a time budget.

This simulation routine was used to create an ensemble of B = 500 simulated overall time budget matrices that mimicked the stochasticity attributable to a reasonable approximation of the measurement error of the sensor. Stored as a tensor with replication on the last axis, the variance of the ensemble of simulations could then be easily calculated for each combination of cow index and behavioral axis. If the underlying simulation strategy is a reasonable representation of the noise in the sensor, then these variance terms will then serve as a sufficient approximation of the relative uncertainty in each data point. We propose that that this information can then be incorporated into the calculation of dissimilarity estimates by serving as penalty terms in the calculation of an ensemble-weighted distance estimator defined in Equation (3).
(3)$$D_{EW}\left(P,Q\right)=\overset{\;}{\underset{k}{\sum }}\frac{{\left(P_{obs}\left(k\right)-Q_{obs}\left(k\right)\right)}^{2}}{\sigma_{P^{*}\left(k\right)}^{2}+\sigma_{Q^{*}\left(k\right)}^{2}}\;$$
$$\sigma_{P^{*}\left(k\right)}^{2}=Variance\;of\;ensemble\;of\;simulated\;values\;for\;Cow\;A\;for\;behavior\;k\;$$
$$\sigma_{Q^{*}\left(k\right)}^{2}=Variance\;of\;ensemble\;of\;simulated\;values\;for\;Cow\;Q\;for\;behavior\;k$$

The rescaling strategy employed in our proposed dissimilarity estimator is strongly inspired by traditional analysis of variance (ANOVA) techniques, thereby providing several insights into its anticipated behavior. First, because the simulations were generated using the multinomial or one of its analogs, we can infer that these penalty terms will not be homogenous across the domain of support, but should shrink as observations approach the boundary. This will allow the ensemble-weighted distance estimator to emulate the rescaling dynamic achieved with the KL distance; however, rescaling at the extremes of the domain will ultimately be bounded by our simulated measurement error, so as not to exceed the precision of the sensor. Second, because we have here emulated measurement error in our simulation using sampling uncertainty, the central limit theorem will apply [9]. Thus, we can anticipate that as the number of observations per animal increases, the impact of measurement error on our inferences will shrink, allowing progressively more subtle differences between animals to come into resolution. Taking this property to its limit, however, can it be said that with enough observation minutes the differences between cows can be inferred with near certainty? That intuition, of course, is at odds with our characterization of a dairy herd as a complex system, and highlights an additional stochastic element that must be accommodated—the behavioral plasticity of the cows themselves in response to changes in the production environment [4].

Given the extended observation window of this particular data set, it would be possible to recalculate time budget conditional on the day of observation, and then use the variance in daily time budget along each behavioral axis as a penalty term. Such estimates would collectively reflect heterogeneity in variance attributed to domain constraints, measurement error, and behavioral plasticity. Such an approach would not, however, be feasible for datasets collected over shorter time intervals with fewer replications, or in applications with behavioral responses where there is no clear hierarchy in the temporal structure of the same. We therefore propose that our stochastic simulation model can be extended to also provide a generalizable means to approximate the uncertainty of the underlying behavioral signal.

As before, the measurement error was simulated by redrawing samples at the finest temporal granularity provided by the sensor. Prior to compression along the temporal axis, however, a random subsample of observations days was selected across all cows, and only these values were used to calculate the simulated overall time budget. If all cows demonstrated comparable levels of consistency in their daily time budgets, then reducing the effective sample size of our simulated data sets through a subsampling routine would increase the ensemble variance estimates. This, in turn, would make our approximation of measurement error hyper-conservative, but this increase would be uniform across all cows. If, instead, some cows were less consistent in their time budgets across days, then the sampling error imposed by the subsampling routine would be greater, resulting in a larger ensemble variance estimate. Thus, we would expect a stronger penalty to be applied to cows who demonstrated greater plasticity in their behavioral responses to both transient and persistent changes in the production environment. For small datasets with a limited number of replications, the number of subsamples could be set quite close to the size of the complete sample, and would thus emulate a jackknife approach to variance estimation [9,10,28]. For larger datasets, however, the subsample size could be set smaller, to make the resulting ensemble variance estimates progressively more sensitive to the uncertainty in the underlying behavioral signal.

To evaluate the empirical performance of these dissimilarity estimators, distance matrices were calculated for the 177 cows with complete CowManager time budget records. Euclidean distance and KL Distance were calculated using base R utilities, with speed up options utilizing the *Rfast* package [23,29]. An ensemble-weighted dissimilarity matrix was first calculated using simulated values accounting only for measurement error using the most conservative joint Dirichlet-multinomial sampling scheme, hereafter referred to as noise-penalized distance. A second ensemble-weighted dissimilarity matrix was then calculated using the same sampling scheme for measurement noise but aggregated over a 14-day subsample to account for behavioral plasticity in daily time budgets, hereafter referred to as plasticity-penalized distance. The LIT package provides users a clustering visualization utility, which converts dissimilarity matrices into a dendrogram using the *hclust* utility in base R with default Ward D2 linkage [23], and the generates heatmap visualizations of the resulting clustering results using the *pheatmap* package [30]. Heatmaps were generated on a grid of cluster values from k = 1 … 10 for each of the four dissimilarity estimators, with complete results provided in Appendix A, the results for k = 10 clusters are provided. The LIT package also provides users with a plotting utility to visually contrast the broader patterns between behavioral encodings. Outputs from the clustering utility are passed in to create a contingency matrix generated using *ggplot2* with cells colored by their corresponding cell count [31]. The heatmap visualizations for each encoding are then added to the row and column margins of the contingency matrix using the *ggpubr* package [32], and arranged such that each row cluster in either heatmap matches the order of the contingency matrix reading either up-down or left-to-right, allowing for direct and detailed visual comparison of the discretized behavioral patterns. Comparisons between the noise-penalized and plasticity-penalized encodings are provided.

### 2.3. Improving Tree Pruning Decisions through Simulation

An optimal encoding strategy seeks to minimize the loss of relevant information by retaining as much of the underlying deterministic signal as possible, while hemorrhaging only noise [26]. In a hierarchical clustering framework, this is achieved by pruning the dendrogram built from the dissimilarity matrix at the point where the branches cease to represent differences in the underlying signal. Standard pruning strategies allow users to either: (1) provide a dissimilarity cutoff, below which value all further branches are grouped into the same bin, or (2) extract the first K branches of the tree [9,10]. As with the default Euclidean distance dissimilarity estimator, this approach may be appropriate for datasets with relatively homogenous variance structures. For data drawn from intrinsically heterogenous distributions, however, the branch lengths cannot be directly compared across the domain of support, making globally-defined pruning rules a suboptimal strategy for analysis of time budget data.

More fundamentally, a homogenous pruning strategy may be too simplistic for many PLF sensor datasets, for which the underlying signal often represents a complex composite of behavioral mechanisms that operate at multiple scales. Although some environmental factors might be expected to have an impact on cattle behaviors that are uniform across the herd, other factors might elicit responses that differ in magnitude for different subgroups within the larger population, or even become isolated within smaller social cliques. For example, we might expect the number of times cows are moved each day for milking will place similar constraints on the time left to lie down across all animals, but overstocking with respect to stall spaces might have a much larger magnitude of impact on the lying patterns of subordinate heifers than the more dominant older cows [33]. In such a complex system, we would expect the heterogeneity imposed by the underlying biological signal to differ in scale across the dataset. Subsequently, in attempting to employ a global cutoff decision to encode information for such a dataset, we would always be faced with the difficult decision to either ignore the subtler behavioral patterns present in some branches of the tree, or else allow noise to contaminate our encoding of other branches with intrinsically coarser behavioral patterns.

Although all the components that contribute to the signal in a complex livestock system might be difficult to anticipate *a priori*, we propose that a more dynamic pruning algorithm might still be achieved, by again employing flexible simulation-based approaches to emulate the comparably simpler sources of uncertainty. If each branch of the dendrogram is viewed as a pairwise contrast between two groups of animals, then we need only to determine whether the bifurcation under inspection represents a difference in the underlying signal that can be reliably distinguished from noise. If it can, then the two groups should be split in the final encoding to capture this feature of the data’s distribution. If a branch falls below the intrinsic resolution of the data, however, then the branch may be pruned so that all animals are placed into the same cluster, with no loss of meaningful information. By implementing such a branch-level test recursively, we can gradually work our way down the tree with adaptive locally-defined pruning decisions.

To evaluate the reliability of the behavioral signal encoded at each bifurcation of the tree [34], our branch test utility utilizes two mimicries. The first set of simulations are generated under the alternative hypothesis that assumes a branch contains an underlying deterministic signal that is only partially obscured by stochastic noise. Thus, we can simply repurpose the ensemble of simulated data sets used previously to calculate the ensemble weighted dissimilarity metrics by mimicking the uncertainty in the observed data. The second set of simulations are generated under the null hypothesis that a given branch contains only noise. As the null implies that animals demonstrate equivalent patterns of behavior within the resolution of the sample, this mimicry can be generated quite efficiently using a standard bootstrapping routine [28], wherein time budgets simulated under the alternative are unconditionally resampled from amongst all animals in a given branch. HClustering is then performed independently on each data mimicry in either ensemble, and the first k branches are extracted to create an ensemble of discrete encodings.

Under the alternative hypothesis, a strong signal should produce a robust tree structure such that, even after the addition of simulated noise, the resulting encoding would still closely mirror that of the original observed data. As the stochastic component of a dataset becomes stronger relative to the signal, these bifurcation points will become progressively less stable, and the subsequent encodings less reliably aligned with the original data. When the signal falls below the resolution of the data, the tree structures of the simulated data would then seldom match that of the original data, and so would become poorly distinguished from encodings generated under the null, with no signal component. We propose that mutual information, which can be calculated without any additional distributional assumptions, can be used to quantify the similarity between the observed data and each mimicked dataset, and subsequently used to determine if simulations under the alternative are distinguishable from the null [12]. In our study, a bifurcation was determined to be significant if less than 5% of the MI values calculated for data simulated under the alternative hypothesis fell below the 95th quantile of MI values calculated for data simulated under the null. If a bifurcation was instead deemed insignificant, the branch was pruned and all cows within it assigned to the same cluster in subsequent encodings.

In evaluating the significance of a bifurcation, it seems intuitive that a k = 2 binary encoding should be utilized. For complex systems subject to the influence of multiple competing drivers of behavioral responses, however, a false negative result can occur with this parameterization if the addition of stochastic noise perturbs the order in which two significant mechanisms with similar magnitudes of impact are bifurcated. Such trivial destabilizations of the tree structures can be readily identified in visualizations of the distributions of MI values calculated against simulations under the alternative, as the “flip flopping” between bifurcation points produces clear evidence of multimodality (see Figure 1). To circumvent this issue, the LIT package provides users the option to re-test any bifurcations deemed insignificant, using a binary encoding with a more granular discretization (k > 2). This effectively allows the algorithm to “look down the branch” to absorb any irrelevant flip-flopping between competing signals, thereby preventing spurious over-pruning that would hemorrhage information on significant behavioral patterns from the final encoding.

Full results for the application of our ensemble-cut algorithm to dendrograms generated using each of the four dissimilarity estimators discussed in the previous section, and using both the noise-penalized and plasticity-penalized encodings, are provided in the Appendix A. A summary of results for the application of the ensemble-cut algorithm applied to dendrograms generated from the noise- and plasticity-penalized ensemble-weighted distance metrics, the noise-penalized and plasticity-penalized encodings, respectively, are provided.

### 2.4. An Information Theoretic Framework for Cross-Sensor Inferences

Equipped with an appropriate encoding to discretely represent the heterogeneity in overall time budgets within this herd, and provided with the encoding of longitudinal patterns in parlor entry position from previous work with this data set, a potential question to ask would be: how does a cow’s time budget, which is largely determined by her behaviors in the home pen, relate to her behavior in the milking queue? There are a number of nonparametric and parametric techniques available to evaluate the overall strength of association between two discrete variables, by evaluating the distribution of animals in the joint encoding [26]. There is, however, perhaps greater practical utility in characterizing low and high points within the joint encodings, which would provide more detailed insights into the tradeoffs between specific behavioral patterns recovered from the data streams in these distinct farm contexts. Towards this end, information theory offers a more comprehensive approach to decomposing the stochasticity within discretely encoded variables, and thus may provide a more holistic approach to evaluating both the global and local features of a joint encoding, while employing few structural assumptions [12].

First, to evaluate the strength of the overall relationship between two discretized behavioral responses, the LIT package provides users a permutation-based bivariate testing utility that uses the mutual information estimator to quantify the amount of information entropy that is redundant between the two encodings [7,12]. We can anticipate, however, that the efficacy of this test in recovering significant relationships between the underlying biological signals will be affected by the resolutions of the encodings. Suppose that a single latent biological factor impacts the behavioral responses collected by both PLF data streams, creating informational redundancy between the two encodings. If we cut the trees above the intrinsic magnitude of its impact on a given behavior, its influence may be overlooked and mutual information underestimated. On the other hand, if we prune the tree far below the magnitude of its impact, our inferences can lose power, as bin sizes in the joint encoding become progressively smaller, weakening the empirical estimation of the joint probability distribution and thereby increasing estimation error in the MI estimator. The resolution of our encodings must, therefore, be optimized to match the dynamics of the system, or a false negative result may be returned. To further complicate matters, however, we cannot necessarily assume that the magnitude of impact of a given latent factor will be uniform across behaviors, nor should we expect in a complex farm environment that behaviors will be influenced by a single latent factor.

To overcome this logistical challenge without falling back on dubious *a priori* assumptions, the LIT package implements mutual information-based permutation tests on a grid, varying the cluster resolutions across both behavioral axes [7]. Under the null hypothesis that no significant bivariate relationship exists between data streams, cow ID labels are randomly permuted within each tree, preserving the marginal distribution of the data along each axis, but destroying any latent bivariate relationships. These permuted trees are then cut, and the mutual information of the joint encoding estimated for each combination of cluster counts on the grid. A p-value is then generated by comparing the observed MI value of the joint encoding at each grid point against the corresponding distribution of MI values simulated under the null. Just as a scientist varies the focus of a microscope to bring microbes of different size into resolution, we can expect that geometric features of the joint probability distribution imposed by latent deterministic variables, that vary in scale of impact, will come into and fall out resolution as these meta-parameters are varied across the grid of cluster counts. To help the user visually identify where such features have come into resolution, the LIT package also returns a heatmap visualization of the observed MI value for each grid point that is centered and scaled, relative to the distribution of MI values under the null. For behavioral measurements subject to the influence of multiple biological and environmental factors operating simultaneously, this exhaustive approach to parameterization enables users not only to build a more complete picture of a complex behavioral system, but may also provide insight into the hierarchy of these behavioral responses.

Unfortunately, as the resolution of the encodings is increased, MI estimates not only become less precise, but they may also become less accurate. Bias is introduced when empirical estimates of the joint probability distribution become so granular (i.e.,a high number of bins relative to the total sample size) that regions with low but nonzero probabilities go unsampled. These zero-count bins cause the total entropy calculated from the empirical joint probability distribution to be underestimated which, in turn, causes the relative amount of redundant information to be overestimated. Although the magnitude of this bias is partially dependent on the total sample size, it is also contingent on the structure of the joint probability distribution itself, namely the number of low-probability cells. Given that the joint probability distribution under the null, which is randomly permuted to intentionally remove any nonrandom features in the sample, can be expected to have a more uniform distribution of probability than the observed dataset, we can anticipate that the magnitude of the bias may differ between these two distributions as the sample becomes more granular, preventing MI estimates from being directly comparable. To overcome this issue, the LIT package by default provides entropy estimators based on the Maximum Likelihood frequency estimates, but allows users to select from a range of bias-corrected frequency estimates available in the *entropy* package [35]. Based on the simulation work by Hausser and Strimmer (2009), the JS “shrink” estimator was used in our study to conduct bias-corrected mutual information permutation tests [36].

Not only can the impact of latent factors on behavioral measures differ in magnitude, we can also anticipate that responses may differ in both strength and direction for different subgroups within the herd. Such nonlinear dynamics are easily captured in a model-free MI test, but further inspection of the contingency table is needed to fully characterize such complex bivariate relationships between sensor outputs. If either marginal encoding has roughly the same number of observations in each bin, then the cell counts in the joint contingency table can be directly compared, as under the null we would expect each cells to be equiprobable. For empirically defined encodings, however, bin sizes can vary significantly to better capture the underlying geometry of the univariate data distribution. Such differences in marginal probabilities prevent the raw cell counts from being directly compared. To better identify which cells in an empirically defined joint encoding are driving a significant overall relationship between two data streams, mutual information can be decomposed into pointwise mutual information (PMI) values [37]. The LIT package provides users the option in the *compareEncodings* plotting utility to color cells in the joint contingency table by PMI estimate, to better facilitate direct visual comparisons of the encodings. To further enhance visualizations of the joint probability distribution that significantly differs from expected cell counts under the null, users may also specify a probability threshold above which PMI values should not be displayed, which was determined here by simulating PMI estimates under the null by redrawing from a multinomial distribution using the outer product of the marginal distributions.

Bivariate tree tests were applied to the time budget encodings, using both the noise- and plasticity-penalized dissimilarity metrics, and pruned using the more conservative plasticity-penalized mimicry, against the encoding of parlor entry order data produced using data mechanics clustering from our previous work [7]. A 2:10 × 2:10 grid was used to determine the optimal resolution for the bivariate relationship, with the optimal meta-parameters used to create visualizations of the joint encoding, wherein pointwise mutual information values were used to color cell counts that were significant at the alpha = 0.05 significance level. To further explore latent factors that might explain significant associations between entry position and time budgets, bivariate tree tests and pointwise mutual information tests were also applied separately to the encodings of both PLF data streams and health records.

## 3. Results and Discussion

### 3.1. Improving Empirical Encodings of Overall Time Budget through Simulation

Figure 2 provides a visual comparison of the time budget encodings for the four candidate dissimilarity metrics. In each heatmap visualization, individual cows are arranged along the row axis, and the mutually exclusive behaviors that comprise the overall time budget are ordered along the columns. Each cell within the heatmap is subsequently colored to reflect the proportion of time that a given cow is recorded by the accelerometer system to engage in a specific behavior over the observation window. Few cows dedicated more than half of their time to any one behavioral axis, which is not surprising, given that total lying time in this system is split between the nonactive and rumination axes [33]. Time recorded as eating and time recorded as ruminating were the highest magnitude behavioral axes, but time spent eating demonstrated far greater range and heterogeneity. Time spent nonactive was lower in overall magnitude, but still showed a fair amount of heterogeneity across cows. The active and highly active axes, however, were both quite low in magnitude and generally demonstrated less systematic heterogeneity across the herd. The order of cows along the row axis in each heatmap is determined by the dendrogram calculated for each dissimilarity matrix. The dendrogram can be interpreted as an approximate 2D representation of the distribution of the cows with the 5D multinomial space of the time budget, and thus serves to bring out in the heatmap systematic differences in time budget across the herd. Gaps were added between rows to indicate branches that have been pruned, such that all cows within a given branch received the same discrete value in the final time budget encoding.

A cursory appraisal of all four encodings summarized in Figure 2 reveals that, regardless of the dissimilarity metric utilized, there was a considerable amount of heterogeneity in the distribution of overall time budgets across this herd. Looking more closely at the clustering tree produced from the unweighted Euclidean dissimilarity metric in Figure 2A, we can see that the higher magnitude eating and rumination axis entirely dominated the first handful of bifurcations of the dendrogram. Even for users not accustomed to reading dendrograms, this dynamic is clearly animated by parsing through the grid of heatmap visualizations provided by the *encodePlot* utility (see Appendix A). Heterogeneity in the moderate-magnitude nonactivity appears to have been largely ignored in the first half-dozen bifurcations, with the first 10 clusters extracted from this dendrogram being ultimately quite variable in the nonactivity response. Nor is there clear evidence that either activity axes influenced the first 10 bifurcations of this tree. This dynamic is almost certainly attributable to the lack of intrinsic scaling with this estimator. While a behavioral axis that represents a larger proportion of a cow’s time investments may warrant additional consideration, these results clearly demonstrate that the Euclidean norm does so to nearly the complete exclusion of lower-magnitude behavioral axes that might still convey important ethological information. The Euclidean distance heatmap is also annotated on the row axis with a number of auxiliary data fields for each cow, which included: age (birth date); calving date; an estimate of peak lactation; nutrition supplementation treatment, and health status during the observation window (see Appendix A for details on the encoding of these auxiliary cow attribute variables). A cursory visual inspection reveals that most clusters appear to be fairly homogenous with respect to cow age, tenure in the pen, and feed supplementation status. Sick cows, however, appear to be slightly overrepresented in some groups, namely the smaller branches representing the more extreme time budget tradeoffs.

Looking next at the hierarchical clustering results visualized in Figure 2B, the KL distance seems to have provided a slightly more holistic encoding of the data that better balances the input across the five behavioral axes. Again, extremes in eating and rumination drive the first few bifurcations of the tree structure, but tradeoffs between time spent eating and nonactivity are considered much earlier in the bifurcation decisions within this tree. Some systematic heterogeneity was also revealed across the herd in the high activity axis, despite its lower magnitude. Unfortunately, the KL distance also appears to have over-stratified cows whose time budgets lie at the extremes. In particular, the cows with extremely low time spent eating (clusters 6–8) were divided into clusters that are likely too small and narrowly defined to facilitate cross-sensor inferences in downstream analyses, and thus may obscure important behavioral dynamics in this dataset. The KL distance heatmap is also annotated on the row axis with the variance in observed daily time budgets for each behavioral axis. Given that time budgets have been normalized here and expressed as proportions, the resulting variance terms were quite small in magnitude (less than zero), and so have been re-expressed on a log-scale, where an increasingly negative value represents a smaller relative magnitude of variation. The fact that all five axes ranged over several orders of magnitude in these variance estimates reveals that there was an appreciable amount of variability in the time budgets across days. Visual appraisal revealed very little systematic patterns in this heteroskedasticity across clusters, however, suggesting that differences in relative plasticity in daily time budget observations may be attributed more to the individual than to any specific pattern in overall time budget.

The noise-penalized ensemble-weighted distance, visualized in Figure 2C, displays clustering dynamics that fall somewhere in between the two extremes of Figure 2A,B. Time spent eating and ruminating still dominate bifurcations nearer the root of the tree, as with the unweighted Euclidean distance, but the most extreme tradeoffs between these axes were here pulled off without over-cutting the tree, as with the KL distance. In the later branches of the tree, however, cows with more moderate time budgets are divided with greater input from the nonactive and highly active axes. Although the ensemble-rescaled estimator does appear to have succeeded in curbing the rescaling of dissimilarity estimates at the extremes of the distribution, the noise-penalized ensemble distance did still bifurcate several cows with anomalously high values in the eating, ruminating, and nonactive axes into their own clusters of size n = 1. Although isolating these animals into their own branches will effectively exclude them from cross-sensor inferences in downstream analysis, this encoding may still be appropriate if these datapoints represent authentic outliers that cannot be explained by typical variation in the sensor system. The heatmap was also annotated on the row axis with the ensemble variance terms used to penalize the squared distance estimates. We see that, as anticipated, the magnitude of error in the noise-penalized ensemble variance terms is substantially smaller than the observed variance in observed daily time budgets, confirming that, with so many samples over an extended observation window, measurement error was not contributing substantially to the overall uncertainty in observed time budgets. Closer appraisal of the clear systematic differences in these ensemble variance terms observed across clusters, however, confirms that these penalty terms appear to be effectively mimicking the intrinsic heteroskedasticity in this multinomial sampling space.

Ensemble variances calculated for each cow via the plasticity-penalized simulation routine closely matched (R ≥ 0.99) the variances in observed daily time budget estimates for all five time budget axes, thereby validating the efficacy of the jackknifing routine. Figure 3 directly contrasts the first 10 clusters extracted from the dendrograms generated by the noise- and plasticity-penalized ensemble-weighted distance measures. In this visualization, clusters are numbered in each heatmap from top to bottom, and so directly align with the row and column indices of the contingency table. For example, we can easily confirm from this graphic that the first three cows constituting the first two clusters in the noise-penalized heatmap were the same cows isolated into the third and fourth clusters in the plasticity-penalized heatmaps—a determination that can be easily confirmed by zooming in on this high-definition rendering to compare Cow ID values. Further comparisons revealed that cluster designations for cows with extremely high time spent eating, extremely low time spent ruminating, and relatively low time spent nonactive (clusters 5 and 6 in the noise- and plasticity-penalized encodings respectively) were virtually identical. In the plasticity-penalized dendrogram, the extremely low eating time cluster (cluster 3) shrunk by just a few animals, compared with the noise-penalized encoding (cluster 4). Additionally, after penalizing for behavioral consistency, the cow with the highest time spent nonactive in the sample (cow 6580) was not isolated as an outlier. This bifurcation was instead shifted to the cows with more moderate time budgets (clusters 7–10), serving to better distinguish between cows with relatively high and only moderate times spent eating. The plasticity-penalized dissimilarity estimator was also notably more generous in assigning cows to the cluster characterized by slightly higher rates of rumination, while all other axes remained relatively low (cluster 7), and appeared to place greater emphasis on the nonactive axis to determine the remaining clusters. Despite these differences, both ensemble-weighted dissimilarity metrics succeeded in producing encodings that provide a more holistic and balanced description of this dataset, and ultimately serve to better visualize heterogeneity in the tradeoffs between all five behavioral axes.

### 3.2. Improving Tree Pruning Decisions through Simulation

For all dendrograms pruned using the ensemble of simulations that accounted only for measurement noise, an extremely fine-grained encoding was returned. A total of 39 clusters were returned for the unweighted Euclidean distance, 31 for the KL distance, and 38 clusters for the noise-penalized dissimilarity metric. In Figure 4A, the heatmap visualization of the noise-penalized encodings helps to illustrate just how far down each branch the pruning algorithm was able to penetrate before the signal was lost to simulated measurement error. In fact, amongst the first dozen bifurcations in this dendrogram, the only branch not validated was that which would have isolated the cow with the highest observed time spent eating (cow 63911) into her own branch. This result is not necessarily surprising, given the extended observational period over which sensor records were recorded. With over 1500 min of observation for each cow, even in using a relatively conservative simulation strategy that very likely overestimated the noise intrinsic to this sensor, we should expect by the CLT that the standard error attributable to measurement error would ultimately be quite small after averaging over so many sampled timepoints. Subsequently, these results reinforce that the sensors themselves should impose few limitations on downstream inferences for this dataset, and that inconsistencies in the environment and the animals themselves should be the true limiting factor for the resolution of this encoding.

As expected, the dendrograms pruned with the ensemble of simulations that accounted for both measurement error and longitudinal consistency of the underlying behavioral pattern produced encodings that were far more granular. A total of 13 clusters were returned for the unweighted Euclidean metric, 17 for KL distance, and 14 for both the noise-penalized and the plasticity-penalized dissimilarity metrics. In Figure 4B, the heatmap visualization of these pruning results for the plasticity-penalized dissimilarity metric reveal an encoding that is coarser but ultimately quite well balanced, with the pruning heights modulated to produce cluster sizes that were reasonably uniform across the domain of support. Closer inspection revealed that this final encoding largely matched the order of bifurcations in the original tree, except that this pruning strategy left no animals isolated in anomalous clusters. It should be noted, however, that the granularity of this encoding is not entirely intrinsic to this system, but was dependent on the size of the subsample used to calculate the overall time budget in each simulation. While we can expect cows that were more inconsistent in their daily time budgets to be subjected to a stronger penalty with this estimator due to relatively higher rates of sampling error imposed by the subsampling routine, we can also anticipate that the overall scale of the sampling error imposed on all cows should grow as the size of the subsample is reduced. This would in turn modulate how quickly the underlying behavioral signals would be drowned out by simulated noise within the tree. This suggest that, for larger samples where a greater range of subsample sizes can be utilized, this simulation value can also be treated as a meta-parameter to tune the granularity of the final encoding. Given that the plasticity-penalized mimicry was created for this data set by subsampling only 14 out of 65 observation days, the resolution achieved in the pruned encodings for all four dissimilarity metrics reinforces that this herd was overall fairly consistent in their daily time budgets, and that this data set will support fairly detailed inferences against a strong underlying behavioral pattern.

### 3.3. An Information Theoretic Framework for Cross-Sensor Inferences

Encodings of the overall time budgets produced using both the noise and plasticity-penalized dissimilarity estimators, wherein both were pruned using the more conservative plasticity-penalized ensemble, produced similar behavioral insights when compared against longitudinal patterns in parlor entry positions across the herd. For the bivariate analyses run with encodings for all 177 cows with complete records, highly significant associations with entry order were recovered for both the noise-penalized (*p* = 0.006) and plasticity-penalized (*p* = 0.005) time budget encodings. The bivariate relationship was optimized for both time budget encodings with a five-cluster encodings of entry-order patterns. The noise-penalized encoding produced the strongest associations with entry order, with seven time budget clusters, whereas the plasticity-penalized encoding performed better with a finer encoding of nine clusters, the key difference being the degree of stratification among animals with the most moderate time budgets.

Visualization of the contingency tables for the optimized encodings colored by their PMI estimates revealed that the significant overall association between the two data streams was driven predominantly by animals in the latter half of the milking queue. Figure 5 displays the results for the noise-penalized encoding. We see first that cows that entered consistently at the very rear of the queue (cluster 1) were significantly overrepresented in the time budget cluster, characterized by moderate time spent eating, low time nonactive, and high rates of rumination (cluster 4). Cows that entered nearer the back of the queue (cluster 2), just ahead of the cows that consistently brought up the rear, were also overrepresented in the same time budget cluster—a trend that was statistically significant for the plasticity-penalized encoding, but only marginally significant for the noise-penalized encodings. In fact, very few animals that entered in the front half of the queue were found to have this time budget pattern, with cows entering just behind the leaders being significantly underrepresented in this time budget cluster. One potential interpretation of this pattern might be that, if these cows were prioritizing time investments in rumination, then this strategy may include hanging back towards the later part of the queue, where they may be able to chew their cud while avoiding the more serious contention for parlor entry position. Further analysis that could facilitate visualization of the cyclical patterns in this time budget data would be needed, however, to confirm this suspicion, and will be left for future work.

While this more moderate tradeoff between rumination and nonactivity demonstrated a fairly straightforward and progressive trend across the milking queue, which might readily have been captured by a linear model, more complex dynamics were found for the time budget cluster characterized by extremely low time spent eating and high time spent ruminating and nonactive (cluster 6). Cows that consistently entered at the very end of the queue were significantly underrepresented in this extreme time budget, while the cows that entered just ahead of them were significantly overrepresented. While an extreme tradeoff in eating and ruminating might be explained by issues with sensor placement, that such cows were not evenly dispersed across the herd may instead indicate a biological driver. Health status naturally comes to mind with such an extreme time budget, and indeed several previous studies have reported higher rates of health complications amongst animals in the latter part of the milking queue [38,39,40]. However, health status alone would not necessarily explain the inversion in association pattern between these two adjacent queue groups.

Previous analyses of milk order records have also revealed that, although cows in general tend to be more consistent in their parlor entry order than would be expected in a purely random system, cows at both the front and rear of the herd tend to be particularly persistent their queuing position [38,39,40,41,42]. It remains unclear in analyses of milk order alone, however, to what degree this pattern is attributable to the cows themselves, and any broader behavioral strategies (syndromes) that they may have adopted, and how much is driven by the natural domain constraints intrinsic to this measurement system [7,41]. In older observational studies of movement patterns in cattle, it has been noted that cow herds appear to be “led” from both the front and the rear of the queue [43,44]. One interpretation may then be that sick cows, who cannot maintain a normal time budget, may also be pushed back by competition for entry position in the milking queue, but they cannot be pushed behind this small group of cows that may be “leading from the rear”, effectively serving as a “caboose” for the longer train of animals, as they move between locations to ensure that stragglers are not left behind. Although this behavioral pattern has not been reported in previous experimental studies, it would also not be surprising that such a nonlinear dynamic might be overlooked in analyses relying on linear modeling methods. Indeed, competition between these two behavioral mechanisms, as part of a more complex behavioral system, may explain why relationships between parlor entry position, home pen behaviors, and health status have proven particularly difficult to reliably establish in previous work [5,40,42].

Follow-up bivariate tests with health records confirmed that cows with health complications were indeed overrepresented in the later third of the milking queue (see Appendix A). Mutual information and PMI values did not, however, reveal a significant link between health status and a finer stratification of these late-entering cows. This result, which considers only confirmed cases of acute illness, may however be under-powered, if this behavioral relationship is also influenced by subclinical illnesses not reflected in these health records. More perplexingly, in bivariate analyses with either encoding of overall time budget, cows recorded with acute illness were not found to be significantly overrepresented among the time budget cluster, with extremely low rates of eating. In Figure 6 we can see that sick cows were, in fact, only significantly overrepresented in the time budget cluster characterized by relatively low time spent eating, moderate nonactivity, and elevated rates of high activity—an association that was significant for both the noise and plasticity-penalized time budget encodings. Interestingly, in repeating the bivariate analyses using an encoding of entry-order patterns fit only to animals with no recorded health events, the same time budget cluster with elevated rates of illness was also found to be overrepresented amongst cows entering near the end of the queue, just in front of the “caboose cows” (see Figure 7). Conversely, in this analysis, absent animals with clinical disease, and animals entering at the very rear of the herd were shown to be overrepresented in the time budget cluster that was perhaps best-characterized as demonstrating the most balanced time investments across all five behavioral axes, whereas cows entering just ahead of the caboose cows were underrepresented in this moderate time budget cluster. These results may add weight to the suspicion that ambiguities between clinical and subclinical illness may be obscuring the role of latent health status as at least one key biological link between home pen and milking queue behavior.

## 4. Conclusions

Time budgets provide a convenient and intuitive means of quantitatively summarizing the behavioral tradeoffs of animals, but multinomial-distributed data present a number of analytical challenges. The results of this analytical case study have highlighted how a novel simulation-based approach may be employed to simultaneously accommodate both the codependency structures fundamental to multivariate-distributed data formats and the complex multi-faceted sources of measurement uncertainty that may be encountered across a broader range of PLF data streams. While such simulations may be more computationally expensive than closed-form estimators, we have demonstrated that an ensemble of data mimicries can be efficiently repurposed throughout the analytical pipeline to improve not only the visualization of these behavioral tradeoffs, but also the compression of such information into robust empirically-defined discrete encodings. It should be noted, however, that the utility of these novel clustering techniques is not restricted to time budget data. The ensemble-penalized dissimilarity estimator and ensemble-cut algorithm that we have introduced in this case study are both fundamentally nonparametric. This means that their implementation is in no way intrinsically restricted to any particular class of data. Subsequently, the choices that a user makes in constructing an appropriate error simulation model are restricted only by their own creativity, allowing this analytical framework to be easily generalized to a much wider array of PLF data streams, and the wider array of complex error structures that they have to offer.

Additionally, while discrete data is typically seen as an impediment to statistical analysis in most model-based approaches, we hope that this analytical case study has served to demonstrate the comparable ease with which insights may be extracted from encoded data when an information theoretic approach is employed. For large, structurally complex, and often informationally redundant PLF data streams, an efficient encoding may be far easier to achieve than a comprehensive model that can fully accommodate the temporal dynamics of behavioral responses in complex farm environments. This may be especially true for data sets where all the factors driving such behavioral responses are not measurable. By avoiding entirely any form of least-squared optimization utilized in most model-based approaches, we have shown that an entirely model-free approach is able to recover nonlinear dynamics between entry order and overall time budget, which likely would have been overlooked if an assumption of linearity had been employed. Although more formal model-based inferences may be warranted for further analysis of the underlying causes of this relationship, the exploratory data analysis tools provided by the LIT pipeline have undoubtably served to create a more comprehensive picture of the complex behavioral dynamics hiding within these two under-utilized data streams.

## Figures and Tables

**Figure 1 sensors-22-00001-f001:**
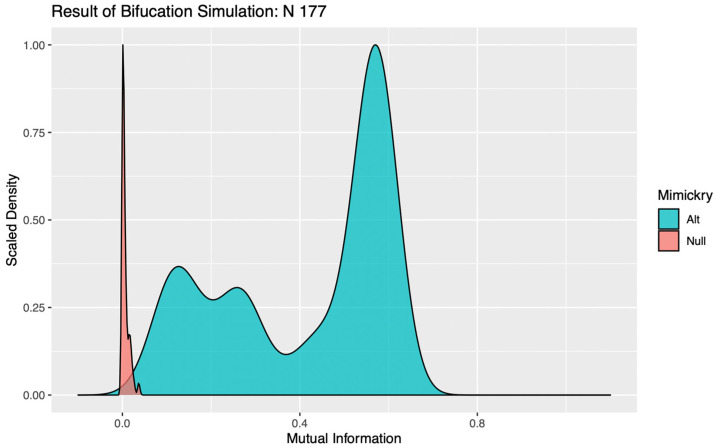
Visualization of the test–branch results for the first bifurcation of the Euclidean distance time budget dendrogram, cut using the noise-penalized ensemble of data mimicries. In simulations under the alternative hypothesis, the addition of noise intended to mimic measurement error has destabilized the tree, causing it to “flip-flop” between first isolating cows with more moderate time budgets, and animals at the two extremes of the tradeoff between eating and ruminating. Although both branches are distinguishable from measurement errors, this ambiguity in bifurcation order has produced bimodality in the distribution of mutual information estimates against the encoding for the observed data. Retesting with more clusters allows the algorithm to “look down the branch” to produce better separation between encodings under the null and the alternative, and thereby avoid spurious over-pruning.

**Figure 2 sensors-22-00001-f002:**
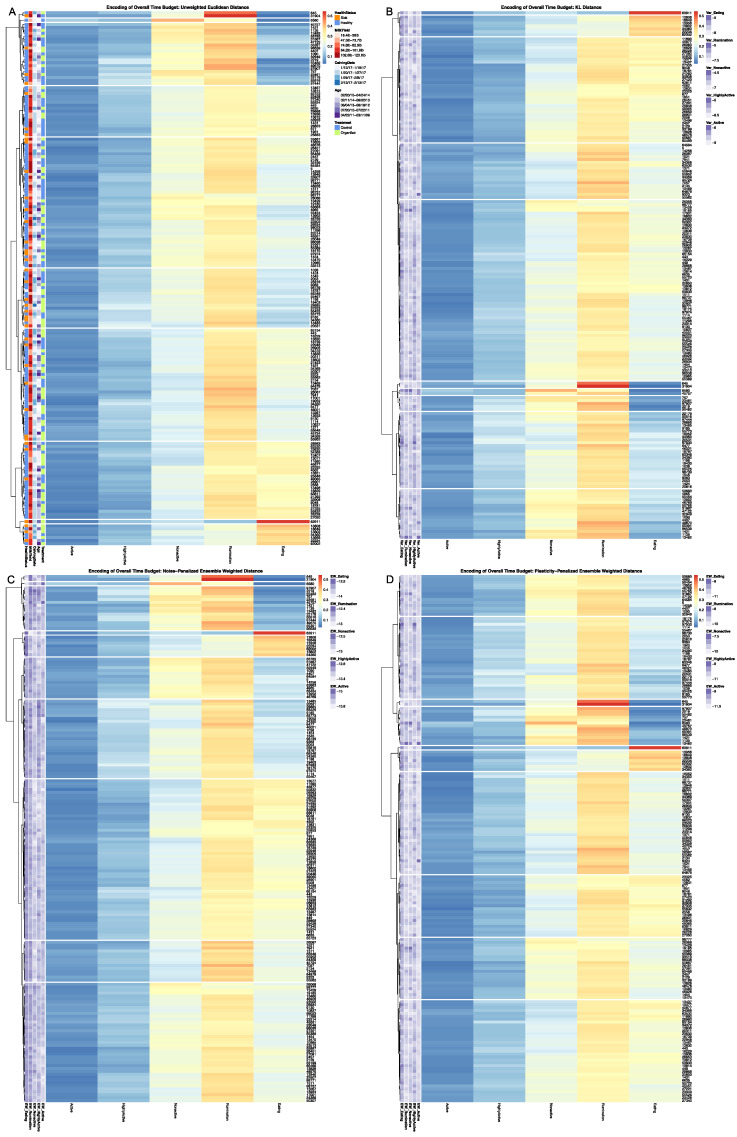
Comparison of overall time budget encodings derived from different dissimilarity metrics. In each heatmap cows are arranged along the row axis, and the mutually exclusive behaviors along the column axis, such that each cell is colored to represent the proportion of time that a given cow is recorded engaging in a specific behavior. Row gaps have been added within each heatmap to reflect the first 10 branches of the corresponding dendrogram, which here are numerically indexed reading from top to bottom (**A**) Euclidean norm encoding with row annotations representing cow-level attributes. (**B**) KL Divergence encoding with row annotations representing log-scaled variance in observed daily time budgets. (**C**) Noise-penalized ensemble-weighted Euclidean distance encoding with row annotations representing the log-scaled ensemble variances. (**D**) Plasticity-penalized ensemble-weighted Euclidean distance encoding with row annotations representing the log-scaled ensemble variances. See Appendix A for full-scale versions of these images.

**Figure 3 sensors-22-00001-f003:**
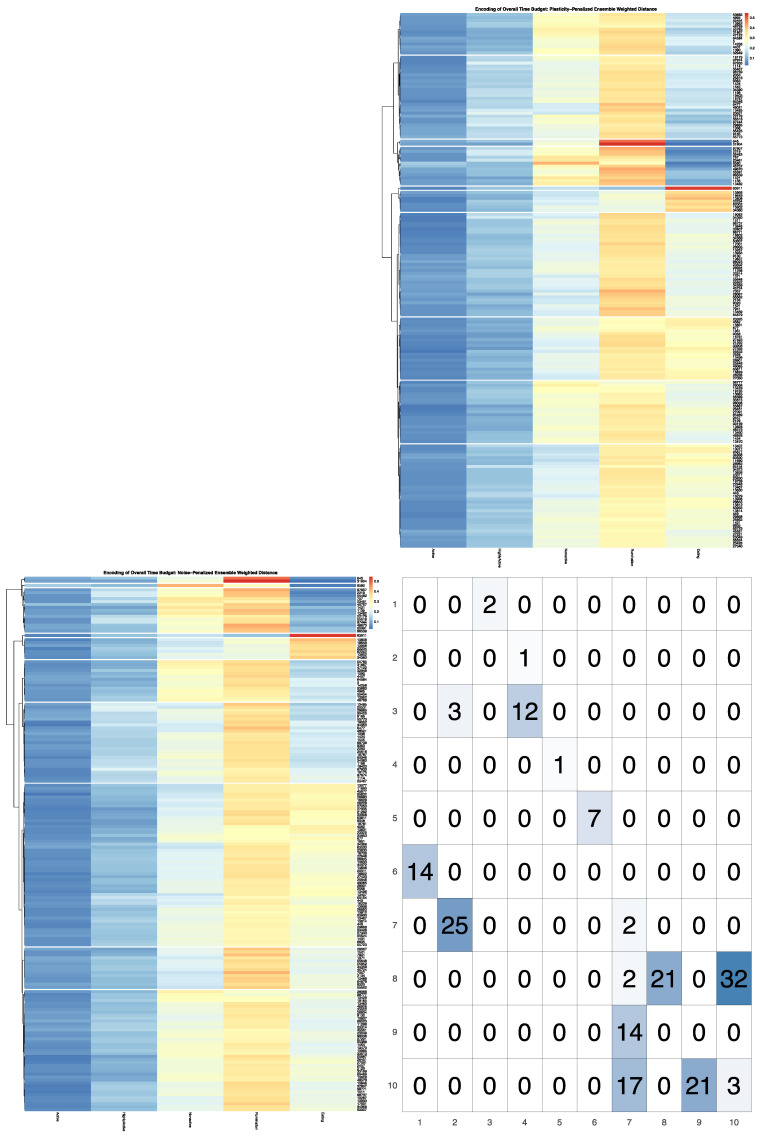
Visualization produced using the *compareEncoding* utility. The noise-penalized encoding is represented on the row axis and the plasticity-penalized encoding is represented on the column axis. Clusters in either heatmap are numbered from top to bottom, and so align directly with the corresponding row and column margins of the contingency table reading up-down and left-right respectively. Cell counts show that these two encodings are quite similar at the extremes of the time budget distribution, but differ slightly in cutoffs amongst the more moderate time budget clusters. See Appendix A for larger versions of these images.

**Figure 4 sensors-22-00001-f004:**
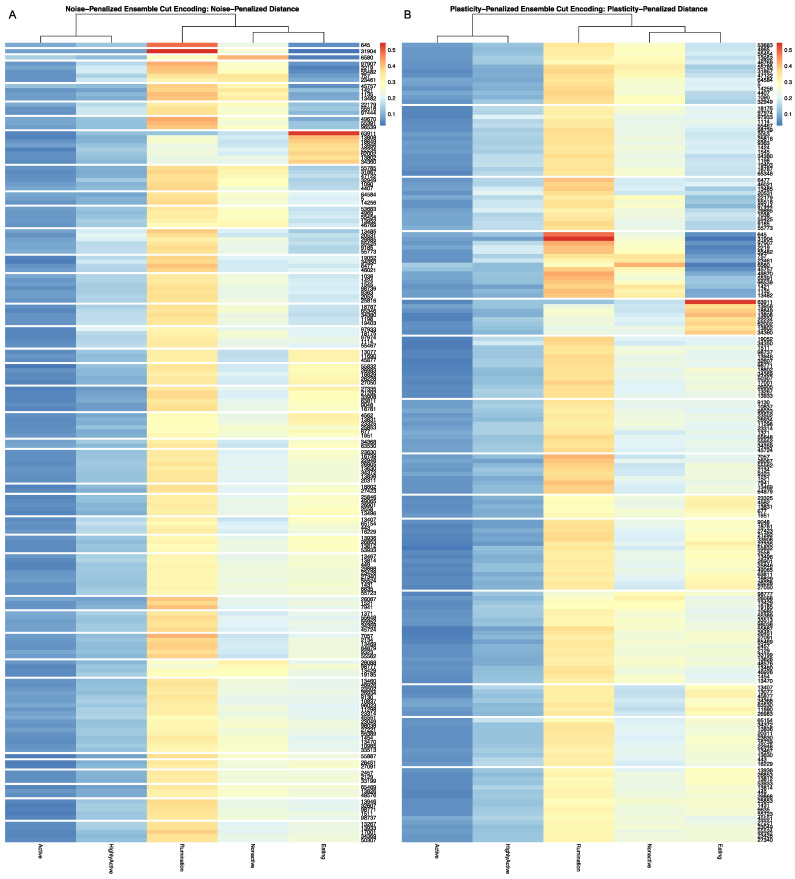
Encodings produced by the cutreeEnsemble algorithm. (**A**) Dendrogram produced by the noise-penalized ensemble weighted dissimilarity metric cut using the noise-penalized data mimicry. The extremely fine encoding with 38 stochastically validated clusters demonstrates that, with so many recorded observations over this extended observation window, the accuracy of the sensor itself should impose few constraints on our behavioral inferences. (**B**) Dendrogram produced by the plasticity-penalized ensemble weighted dissimilarity metric cut using the plasticity-penalized data mimicry. A courser encoding is returned when uncertainty in time budget observations attributable to the behavioral plasticity of the animal itself is taken into consideration.

**Figure 5 sensors-22-00001-f005:**
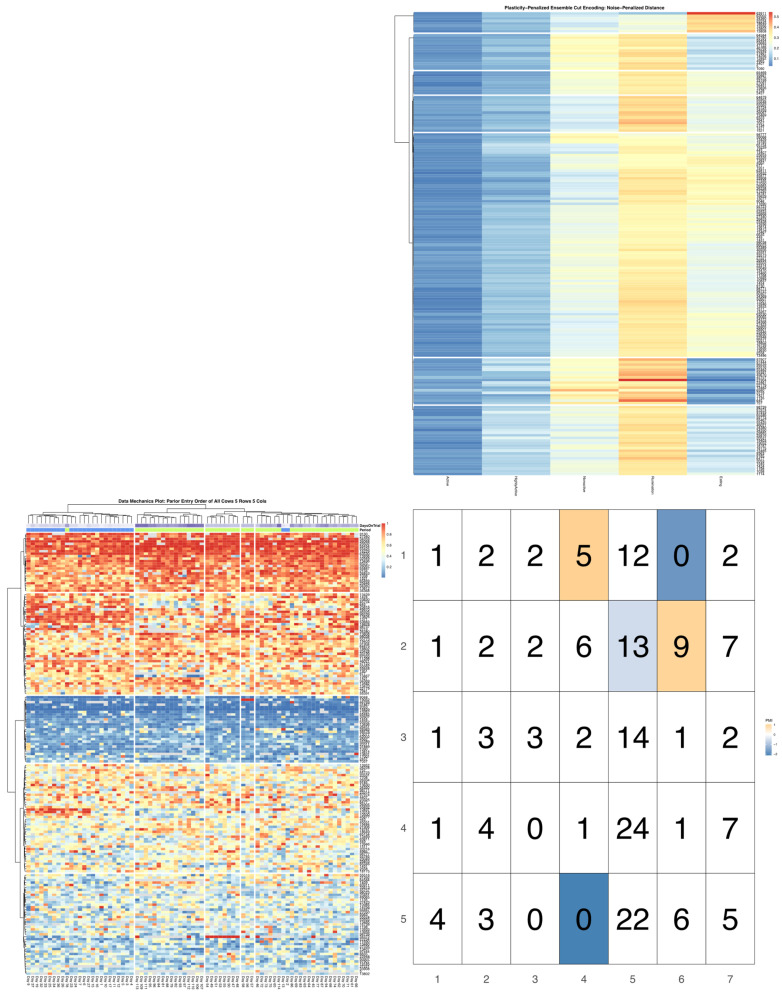
Visualization produced using the *compareEncoding* utility with cells colored by pointwise mutual information estimates significant at the alpha = 0.05 significance level after simulations using multinomial resampling. Data mechanics encoding of parlor entry position is presented to the row margin of the contingency table, wherein the heatmap contains row annotations representing days on trial and the observation period, such that the pen period corresponds with the observation window of the overall time budget. The noise-penalized encoding of overall time budget is represented on the column axis of the contingency table. Pointwise mutual information values reveal that the significant MI test between these two encodings is driven predominantly by behavioral patterns amongst cows in the latter half of the milking queue.

**Figure 6 sensors-22-00001-f006:**
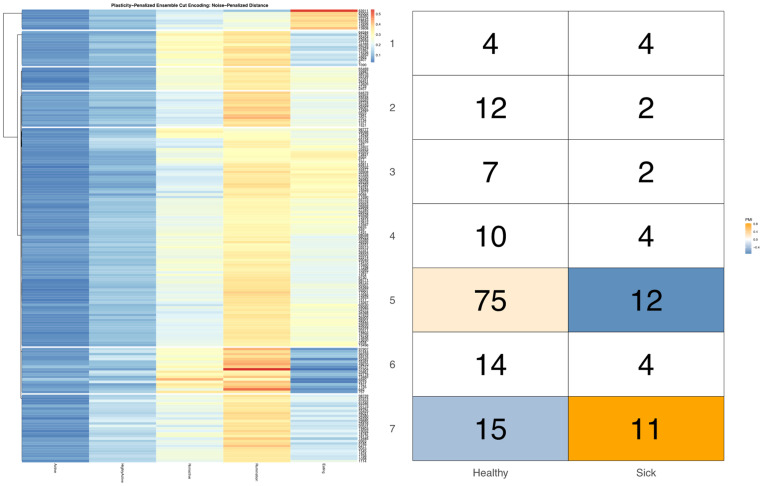
Visualization produced using the *compareEncoding* utility with cells colored by pointwise mutual information estimates significant at the alpha = 0.05 significance level after simulations using multinomial resampling. Cows with the most moderate time budgets were overrepresented among animals with no recorded health events, while sick cows were overrepresented in the overall time budget cluster characterized by relatively low time spend eating and low-to-moderate amounts of time spent nonactive.

**Figure 7 sensors-22-00001-f007:**
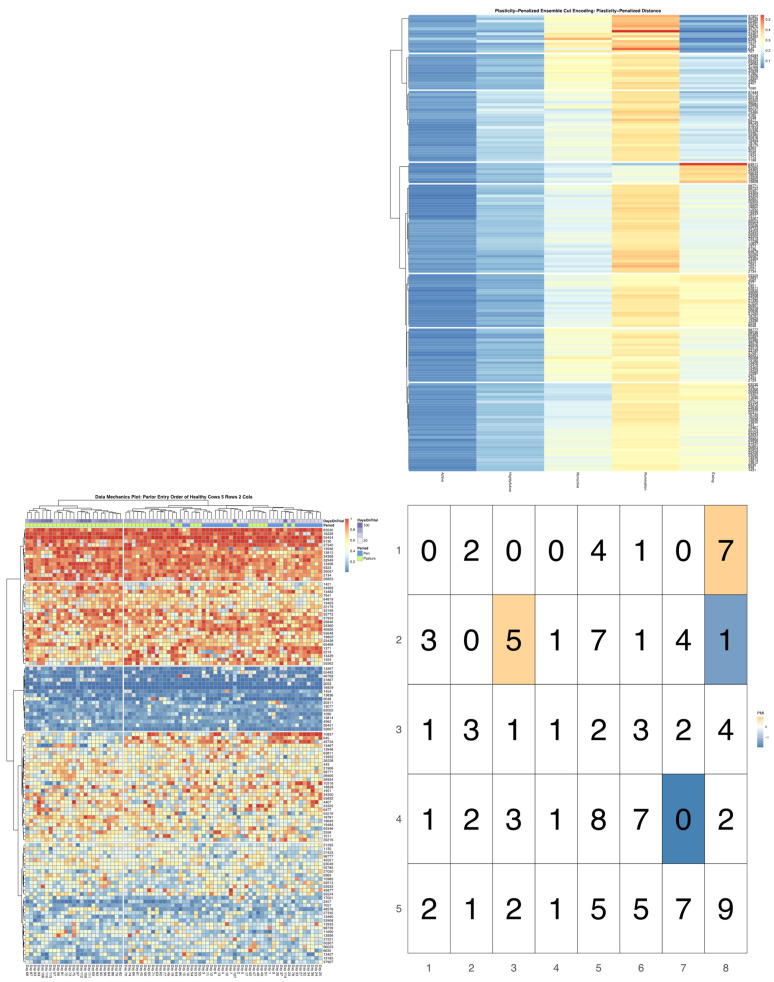
Visualization produced using the *compareEncoding* utility with cells colored by pointwise mutual information estimates significant at the alpha = 0.05 significance level after simulations using multinomial resampling. Data mechanics encoding of time budget data using only cows with no recorded health events is represented on the row axis, and the plasticity-penalized encoding of overall time budget is represented on the column axis. Among cows with no acute illness, cows at the very end of the queue are now overrepresented in the time budget cluster characterized with fairly high time spent eating (cluster 8). Cows entering just ahead of them are not only underrepresented in this high eating time cluster, but are also overrepresented in the cluster with relatively low eating time cluster with low-to-moderate nonactivity (cluster 3) that was independently associated with higher rates of clinical illness.

## Data Availability

The datasets generated for this study are available on request to the corresponding author.

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
