# Peer review of "Livestock Informatics Toolkit: A Case Study in Visually Characterizing Complex Behavioral Patterns across Multiple Sensor Platforms, Using Novel Unsupervised Machine Learning and Information Theoretic Approaches"

_sensors, 2021, doi:10.3390/s22010001_

Round 1

Reviewer 1 Report

The topic of the paper is interesting. It raises many aspects of the new use of sensor data in dairy cows, but in my opinion the content is presented in a way that is too long. What I find missing from the paper is how the results of analyzing data streams can improve the work of the breeder. What new useful and practical information to provide? This has been too poorly articulated.
I think that the structure of the article should be more clearly and better elaborated.
Introduction needs a better flow to the topic and a better indication of the utilitarian aspect of the research. The methodology needs improvement in description and shortening or moved some parts to an appendix. Some content of the article should be moved between chapters to achieve scientific clarity. Separate also the results from the discussion. As it stands, the discussion with the results of other authors is inadequate or it is missing.
More comments in attached file

Reviewer 2 Report

Overview

This paper deals with the complex and pertinent issue of combining different data streams from multiple sensing devices and data creating processes. It utilises a large dataset from a large herd of dairy cows over many days. The paper takes the reader through different stages and options to analyse the data and to particularly group cows, and their data classes,  into different data groupings.  

I found it a challenging read. I accept the core scienceargument, and broadly accept the statistical arguments, but I frequently found it difficult to follow the detail of the arguments . Some novel statistical terms and methods made it hard to follow. Of the four groupings of potential readers – the animal science driven precision livestock scientists – the dairy cow animal scientists – bio-statisticians – and the technology developers (including the software/analytics), I think all groups would be challenged (in different ways) by the descriptions in the text, and find it difficult to fully follow the complex graphics, the within-figure labels, and their legends/captions.

My recommendation is to accept the paper with minor changes, but actually ask for a full review of the text and figures. There are a number of typos, some grammatical issues and I reflect that there are needs to better explain sections, and I believe therefore many more not identified.  I think it would be valuable to go through the paper and where possible simplify text,  and add short elements of explanation or relevance to try and bring those who are not specialist bio-mathematicians in this arena into greater understanding.

The graphics are a key part of the paper because its basis is vizualisation – the core graphic/plot will by necessity be large, but the text that goes with them, either on simple axes, or the tables within the figures is often tiny in font size and difficult to read without zooming text. These need some work to make them accessible both from a basic readability but often to improve accessibility re their meaning/understanding.  Breaking up the Figures, removing the contingency tables into true Tables and providing more conventional and well-labelled legends, table row and column headings would all aid readability.

Neither the ‘Supplementary’ or the Github LIT were accessible – the url of both gave ‘404’ errors from within the pdf version, but also the Supplementary link on the editorial yielded a zip file for which the component files were not readable by any software I had. These need checking.

I have down-graded overall merit because of the challenges in the text, and the inadequacy of  figure presentation. These are a core part of the paper and instead of being a strength, with  poor labelling and the need for clearer self-standing explanations, they are a missed opportunity.  

Minor comments

L72/73 repeats

L85 correct et al x 2

L261 “the behavioral plasticity” , this  sentence/phrase might be better with a reference?

297 typo penalized

299 nonstationary – could be non-stationarity

369 correct to a priori

L476 ‘…by both sensors,’ strictly the sensor is the proximate equipment that collects the data, which needs to be communicated,processed…..?sensing systems?

530 delete ‘the’ before MI test

531 an either ?

533 we’d ..we would

Figure 1 As noted earlier, these figures need review. Using 1A as an example, the dendrogram is unlabelled, the first five columns are coded by the legend at furthest right whilst the main 5 coloured columns by the unmarked scale  0 to 0.5 with no indication what the units are – Euclidean distance…Age label actually refers to ‘date of birth’ and of the column of long numbers are individual cow IDs. All these elements can be deciphered but making at least one of the figures could have column and scale legends and a more readable font size. Trying to get all 4 on a single page looks a challenge and Figure 4 looks a better model with just 2 (though this still needs a large font for text within Figure and there appears no good reason for the bottom axis label ‘Active’ etc  to be vertical when a standard horizontal text would work well enough).

L587 review and rewrite

Figure 2 Caption. The contingency table and the row/columns 1-10 for each. There is an opportunity here to more clearly explain how the two dendrogram plots with their 10 clusters in each create numbers/counts within the contingency table. Explaining for cluster 10 in one plot, and cluster 10 in plot 2 creates the 3 at the bottom-most cell in the contingency table would be a good way to help the reader through  this figure. Again the font size seems inadequate on the two main plots.

Figure 3 – the Mimickry label of Null, Alt presumably refers to the two different plot lines. No colours, dashs to help show which line is which.

L645 typo

L651 typo

L726 ‘caboose’ cows. Is this your term or common US farmer slang? Whilst descriptive, not sure it is appropriate for an international journal

Figure 6: the two number histogram is not described well. The table of counts in the cell table could be created into a better articulated table with heading for columns and rows and a totals for each column that  replace the simplistic histogram. This table could be an actual Table and be separated from the plot graphic enabling the text and font size to be larger and the various column headings to be laid out more conventionally and larger. Cow IDs themselves; are they needed throughout the Figures? Its important to know each row is a different cow, but is the actual ID needed?

L818 we’ve – we have

819 typo

823 not severed ..served?

Reviewer 3 Report

Dear authors,

The reviewed article entitled ‘Livestock Informatics Toolkit: Visually characterizing complex behavioral patterns across multiple sensor platforms using information theoretic approaches and unsupervised machine learning’ is quite interesting and describes a topic that is relevant for Sensors. The manuscript is really well written and good organized. This paper has practical and scientific value. The references are appropriate and used correctly in the manuscript. The detailed M&M gives a strong added value to the manuscript. Authors have obtained interesting results, which are presented generally, in a clear and transparent way. The conclusions are well described and clear and consistent with the evidence and arguments. 

Author Response

We thank the reviewer for their positive comments. 

Reviewer 4 Report

The work is very interesting and scientifically rigorous. The strengths lie in the fact that it proposes a method of analysing and representing big data that takes particular account of the possibility of mimicking the error structure of the data. I must say that I find the methodological part a bit too long while I really appreciated the application part and I would have been interested in being reported in the main part of the article some results that remained in the supplementary section (such as the results for the health status of the animals). But I do not ask for that now (otherwise the paper would be too long).

The weaknesses are not in the method but in some aspects mainly related to the visualisation of results. The figures, although very interesting, are not easy to read, especially in the paper version of the document. It would be necessary to enlarge them using a whole page for each figure. The meaning of the gradient scale from 0.1 to 0.5 (blue to red) should be specified for all dendrograms: does this represent the overall time budget (estimated percentage of time spent on the corresponding behavioural axes as explained in the methods section)?

In the figures (e.g. Figure 1.B) the variability of the 5 behaviours is expressed in a scale with negative values (e.g. for var_eating from -8 to -5). It is not easy to understand the meaning of variability expressed in negative values (they are a log-scale evaluation). A quick explanation of this negative variability should be given somewhere in the text. I understand that these are penalties but perhaps a description of what they mean would make the picture clearer. Is the variability greater when in absolute value this quantity is higher?

Sometimes the figures also fail to describe the part relating to the auxiliary variables: for example, figure 5 shows a ‘period’ variable which has never been referenced in the text (I later saw in the supplementary material that it is a pen/pasture variable). Each element in the figures should be described (either in the indentation of the figure or in the results).  

Figure 3 is referred in methodological part of the article whereas it was reported in the results part where it is never mentioned. In addition, this black-and-white version of the figure is not at all clear, whereas there is a coloured version in the supplementary material (BifucationViz.pdf in EnsembleCut directory) which is more clear.

Author Response

We thank the reviewer for their constructive comments. Please see the attachment.

Round 2

Reviewer 1 Report

Dear Authors,

Thank you for the changes made and explanations given. I am satisfied with them. I have no further comments.

Kind regards